# The Potential of the Fibronectin Inhibitor Arg-Gly-Asp-Ser in the Development of Therapies for Glioblastoma

**DOI:** 10.3390/ijms25094910

**Published:** 2024-04-30

**Authors:** Maria L. Castro-Ribeiro, Vânia I. B. Castro, Joana Vieira de Castro, Ricardo A. Pires, Rui L. Reis, Bruno M. Costa, Helena Ferreira, Nuno M. Neves

**Affiliations:** 13B’s Research Group, I3Bs—Research Institute on Biomaterials, Biodegradables and Biomimetics, University of Minho, Headquarters of the European Institute of Excellence on Tissue Engineering and Regenerative Medicine, AvePark, Parque de Ciência e Tecnologia, Zona Industrial da Gandra, 4805-017 Guimarães, Portugal; norcrib@gmail.com (M.L.C.-R.); vaniacastro@i3bs.uminho.pt (V.I.B.C.); joana.castro@i3bs.uminho.pt (J.V.d.C.); rpires@i3bs.uminho.pt (R.A.P.); rgreis@i3bs.uminho.pt (R.L.R.); 2ICVS/3B’s—PT Government Associate Laboratory, 4710-057/4805-017 Braga/Guimarães, Portugal; bfmcosta@med.uminho.pt; 3Life and Health Sciences Research Institute (ICVS), School of Medicine, Campus Gualtar, University of Minho, 4710-057 Braga, Portugal

**Keywords:** glioblastoma, fibronectin inhibitor, RGDS-functionalized hyaluronic acid hydrogel, liposomes, doxorubicin

## Abstract

Glioblastoma (GBM) is the most lethal and common malignant primary brain tumor in adults. An important feature that supports GBM aggressiveness is the unique composition of its extracellular matrix (ECM). Particularly, fibronectin plays an important role in cancer cell adhesion, differentiation, proliferation, and chemoresistance. Thus, herein, a hydrogel with mechanical properties compatible with the brain and the ability to disrupt the dynamic and reciprocal interaction between fibronectin and tumor cells was produced. High-molecular-weight hyaluronic acid (HMW-HA) functionalized with the inhibitory fibronectin peptide Arg-Gly-Asp-Ser (RGDS) was used to produce the polymeric matrix. Liposomes encapsulating doxorubicin (DOX) were also included in the hydrogel to kill GBM cells. The resulting hydrogel containing liposomes with therapeutic DOX concentrations presented rheological properties like a healthy brain. In vitro assays demonstrated that unmodified HMW-HA hydrogels only caused GBM cell killing after DOX incorporation. Conversely, RGDS-functionalized hydrogels displayed per se cytotoxicity. As GBM cells produce several proteolytic enzymes capable of disrupting the peptide–HA bond, we selected MMP-2 to illustrate this phenomenon. Therefore, RGDS internalization can induce GBM cell apoptosis. Importantly, RGDS-functionalized hydrogel incorporating DOX efficiently damaged GBM cells without affecting astrocyte viability, proving its safety. Overall, the results demonstrate the potential of the RGDS-functionalized hydrogel to develop safe and effective GBM treatments.

## 1. Introduction

Glioblastoma (GBM) is the most lethal and frequent malignant primary brain tumor in adults. Most patients do not survive more than 12 to 15 months after diagnosis [1,2]. Indeed, long-term survival is practically inexistent (a five-year survival rate of ≈3%) [3]. The GBM standard treatment consists of surgery for maximum tumor resection, followed by radiotherapy (RT) and oral administration of a chemotherapeutic agent, namely, temozolomide (TMZ) [4]. However, the complex nature of this high-grade malignant tumor (e.g., high invasiveness and difficulty in delineating its limits) and concerns about the damage to healthy brain tissues make complete surgical resection difficult. Additionally, the remaining cancer cells tend to proliferate during the post-surgery patient recovery period before initiating chemoradiotherapy. However, this treatment modality may potentially be ineffective. Indeed, some cancer cells do not receive the required lethal dose of RT due to, for example, challenges in accurately targeting the tumor, inequitable radiation distribution, and hypoxia. Additionally, some GBM locations have an intact blood–brain barrier (BBB), which reduces the local bioavailability of chemotherapeutic agents [5,6,7]. The existence of proliferating GBM stem cells also contributes to the rapid progression of recurrent GBM because they are more resistant to standard therapy [8,9]. The recurrent GBM has higher genetic variability, complexity, and treatment resistance than the original tumor. Thus, recurrent GBM frequently reduces the survival time of the patients to less than 6 months [10].

To improve the clinical outcomes and consider the benefits of local delivery (e.g., to overcome low BBB-associated permeation, to reduce systemic side effects, and to prevent systemic drug clearance and/or degradation), alternative routes of administration are being investigated, such as direct injection [11,12], convection-enhanced delivery [13,14], and implantable drug-impregnated polymers [15,16,17]. Despite some promising results with local treatments, only the Gliadel^®^ wafer was approved by the FDA [4]. Nonetheless, its use is debatable because of the uncertain survival benefits and potential adverse effects [18]. Like the Gliadel^®^ wafer, local formulations are typically composed of synthetic polymers that serve only as physical depots for drug release [15,16,19]. However, the use of natural polymers can be beneficial. Specifically, high-molecular-weight hyaluronic acid (HMW-HA) emerges as a promising material to be used in the engineering of therapeutic strategies for GBM local treatment. HA is a key structural component of the brain’s extracellular matrix (ECM) and has the potential to aid in brain repair [20,21,22].

In addition to alternative routes of administration, new therapeutic strategies have been continuously investigated. Some approaches take advantage of differences between the tumor microenvironment (TME) and normal brain tissue by developing affinity ligands, such as antibodies and peptides, for specific up-regulated components in the malignant tissue. This feature enables, for instance, targeted delivery of imaging and therapeutic agents within GBM [23,24]. Other studies aim to modify the TME composition by employing strategies like RNA interference to decrease the production of particular components or the use of drugs like dexamethasone to alter the physical or chemical structure of ECM molecules [25,26]. Nonetheless, tumor cells overexpress numerous ECM components and receptors, aiding GBM’s invasiveness and resistance to treatments. Among the ECM components secreted by GBM cells, fibronectin is of particular importance due to its role in tumor cell adhesion, differentiation, proliferation, and chemoresistance [27,28,29,30,31]. As a result, the inhibition of cancer cells’ binding to fibronectin may have therapeutic effects in GBM treatment. Furthermore, the disruption of cell communication and attachment to ECM components could lead to apoptosis by anoikis and sensitize cells to therapeutics [32]. Therefore, inhibitors of GBM cell–ECM interaction can have an important therapeutic role.

In this study, a fibronectin inhibitor, the peptide arginine-glycine-aspartic acid-serine (Arg-Gly-Asp-Ser; RGDS), was used to functionalize HMW-HA and produce a matrix that promotes GBM cell attachment while inhibiting pro-tumorigenic signals. Indeed, while RGD, a peptide widely used to increase cell adhesion to hydrogels, is an integrin recognition motif found in fibronectin, acting as a true agonist of this natural ligand [33], RGDS is an antagonist of fibronectin and, consequently, can avoid its pro-tumoral signaling pathways [34]. Thus, the resulting matrix, presenting a viscoelastic behavior similar to that of brain tissue, enables the desired interaction and attachment of surrounding cancer cells while avoiding the detrimental effects of TME. Large unilamellar vesicles (LUVs) were included in the RGDS-functionalized HMW-HA hydrogel as physical cross-linkers [35] and to encapsulate and enable the sustained release of one of the most powerful chemotherapeutic drugs clinically used, namely, doxorubicin (DOX) [36]. For proof-of-concept regarding the in vitro efficacy of the RGDS-functionalized hydrogel, we used GBML42, a human patient-derived primary GBM cell culture established a few years ago. Consequently, these GBM cells have not been in culture for as many years as the commercially available cell lines (e.g., U87 was established in 1966). Primary cell lines closely replicate the genetic features of tumors, generating data that are more predictive of the real therapeutic value of the developed formulation [37]. Finally, astrocytes, brain cells responsible for a multitude of tasks that contribute to brain health, were selected to evaluate the safety of this approach [38].

## 2. Results

### 2.1. Characterization of LUVs

The size, polydispersity index (PDI), and zeta potential of LUVs prepared with dipalmitoylphosphatidylcholine (DPPC) and incorporating or not DOX are present in Table 1. LUVs with DOX (LUVs+DOX) presented a slightly higher size, PDI, and zeta potential than empty LUVs. The differences were statistically significant for size (*** *p* < 0.001) and PDI (**** *p* < 0.0001). High-performance liquid chromatography (HPLC) analysis demonstrated that LUVs can encapsulate 68.2 ± 13.5 µM of DOX for a phospholipid concentration of 1 mM.

The homogeneous suspension of LUVs (PDI < 0.2) was stable for the tested period (10 days), presenting at the end of the experiment a size of 124.4 ± 5.9 nm, a PDI of 0.111 ± 0.035, and a zeta potential of −3.34 ± 1.63 mV (Appendix A).

LUVs and LUVs+DOX morphologies were analyzed by scanning transmission electron microscopy (STEM). As can be seen in Appendix A, they demonstrated a round shape. Moreover, their size values were in agreement with the dynamic light scattering (DLS) results (Table 1).

The thermograms of LUVs presented a phase-transition temperature (T_m_) of 63.85 ± 1.38 °C and a temperature of onset (T_onset_) of 59.40 ± 1.08 °C (Appendix A). The incorporation of DOX into liposomes led to the presence of a more defined endothermic peak at 51.46 ± 0.35 °C with a T_onset_ of 49.11 ± 0.14 °C (Appendix A).

This set of results showed that LUVs presented the desired properties for biomedical applications, namely, to be encapsulated into hydrogels and to be used in in vitro assays.

### 2.2. HA and RGDS-Functionalized HA Hydrogels Characterization

#### 2.2.1. Nuclear Magnetic Resonance and Attenuated Total Reflection Fourier-Transform Infrared Spectroscopy Analyses

Nuclear magnetic resonance (NMR) and attenuated total reflection Fourier-transform infrared (ATR-FTIR) spectroscopy analyses were performed to assess if the functionalization of HA with the peptide RGDS was successful.

As it was challenging to analyze the peaks of RGDS linked to 1% of the carboxylic groups on HA in the ^1^H-NMR spectrum, its concentration in the hydrogel was significantly increased to facilitate this assessment. The linking of HA to RGDS led to the presence of fingerprints of the unmodified polymer at δ = 3.8–4.4 ppm and δ = 4.9–5.3 ppm (Appendix A), corresponding to the signal of protons in the sugar rings and anomeric protons resonances, respectively. Additionally, the signal at δ = 2.51 ppm corresponds to the protons of the methyl moiety (CH_3_) from the N-acetyl-D-glucosamine unit of HA. The peak at 2.9 ppm corresponds to the proton 13 of Asp, and the signal at 3.4 ppm corresponds to the protons 10 and 3 from Arg and Gly, respectively. The signals of both amino acids (Gly and Arg) suffered, as expected, a small shift when compared to the free peptide (Appendix A; 3.2 ppm for proton 10 and 3.8 ppm for proton 3). The region of δ = 1.6–1.9 ppm corresponds to the protons 9 and 11 from Arg. These signals are not very evident due to a restriction on their mobility after functionalization. The signals of alpha protons (Arg, Asp, and Ser) are collocated under the HA signals.

The ATR-FTIR analyses (Appendix A) demonstrated that functionalized HA presents absorption bands of the glycosaminoglycan spectrum, namely, at 3313 cm^−1^ (O-H group), 2885 cm^−1^ (asymmetric and symmetric stretching vibrations of -CH_2_ group), and 1039 cm^−1^ (C-O-C vibrations) [39,40]. The RGDS-HA spectrum also presented the main characteristic groups of RGDS, namely, the amides I (C = 0 stretching) and II (NH_2_ scissoring vibration of amides in primary amide and from a mixed vibration of N-H bending and C-N stretching in secondary amides) vibration frequency bands at 1630 and 1562 cm^−1^, respectively [41,42].

These results confirmed the successful linking of RGDS to HA, which allowed for the production of the functionalized hydrogel that will be further evaluated in the in vitro assays.

#### 2.2.2. Rheological Properties

To select the HA concentration to use in the following assays, hydrogels were prepared at different concentrations of this natural polymer, namely, 1, 2.5, and 5% (w/v). The elastic shear (G′) and viscous (G″) moduli of the HA hydrogels are presented in Figure 1 and Appendix A. G′ is the elasticity or the elastic energy stored and recovered in the material per deformation cycle, and G″ represents the viscous characteristics of the gel or the energy dissipated during the cycle [43]. The G′ and G″ moduli increased with the concentration of the natural polymer and decreased with the increase in temperature. The concentration of 5% (w/v) was selected to perform the remaining assays since the resulting hydrogels presented mechanical properties similar to the brain tissue [44].

The rheological properties of hydrogels of HA and RGDS-HA incorporating or not LUVs are presented in Figure 2. The values of G′ and G″ are presented in Appendix A. The functionalization of HA decreased the values of G′ and G″. The differences between HA and RGDS-HA hydrogels and between HA+LUVs and RGDS-HA+LUVS hydrogels were statistically significant for G′ (**** *p* < 0.0001 at 25 °C and *** *p* < 0.001 at 37 °C and ^††^
*p* < 0.01 at 25 °C and ^†^
*p* < 0.05 at 37 °C, respectively). Moreover, the differences in mechanical properties of the hydrogel after the addition of 150 µM of LUVs were not statistically significant. This LUV concentration was selected according to the results of the DOX concentration in LUVs. The increase in temperature slightly decreased the mechanical properties of the hydrogels. However, this difference was only statistically significant for HA hydrogels (G′: ^‡‡^
*p* < 0.01).

These results demonstrated that the final formulation (HA-based hydrogel functionalized with the peptide and cross-linked with the LUVs) still has mechanical properties similar to those of brain tissue. Indeed, this demonstrates that the first goal of this study was achieved, namely, to provide a matrix with a viscoelastic behavior compatible with the brain tissue to allow for the desirable interaction and attachment of the surrounding cancer cells without the detrimental effects of TME. [44].

#### 2.2.3. Thermal Properties

The functionalization and the addition of LUVs caused a shift in the endothermic peak of the HA hydrogels (Appendix A and Appendix A). The enthalpy change (∆H) of HA hydrogels was generally higher than that of the RGDS-HA formulations. The ∆H in the endothermic peak of HA was significantly different from the values of RGDS-HA (* *p* < 0.05). Additionally, HA and RGDS-HA have an endothermic T_onset_ at 26.56 ± 2.75 °C and 23.37 ± 0.20 °C, respectively (Appendix A). HA with empty LUVs and with LUVs incorporating DOX presented an endothermic T_onset_ at 26.91 ± 4.88 °C and 25.07 ± 1.62 °C, respectively (Appendix A). RGDS-HA with empty LUVs or LUVs incorporating DOX thermograms showed endothermic T_onset_ at 28.49 ± 2.38 °C and 21.35 ± 2.14 °C, respectively (Appendix A).

These results demonstrated the stability of the different formulations at the temperature used in the in vitro assays (37 °C).

#### 2.2.4. DOX Release by HA and RGDS-Functionalized HA Hydrogels

In this assay, 1 mM was the concentration selected for LUVs to obtain measurable concentrations of DOX. As can be observed in Figure 3, RGDS-HA hydrogels incorporating liposomes released a higher amount of DOX over time than hydrogels with a similar composition but using the unmodified polymer. Consequently, DOX can be released by the hydrogels to damage the cells that are in contact with them or have no contact with them.

### 2.3. Biological Performance

#### 2.3.1. DOX IC_50_ Determination

Appendix A presents the dose–response curves obtained in this experiment, and Table 2 shows the IC_50_ (drug concentration required to kill 50% of cells) values at different time points. As can be observed, the IC_50_ of DOX for GBML42 cells decreased significantly from day 1 to day 2. Moreover, this assay demonstrated that the final formulations have relevant therapeutic concentrations of DOX.

#### 2.3.2. Metabolic Activity of GBML42 Cells in the Presence of LUVs with and without DOX

The cells’ metabolic activity in the presence of LUVs is presented in Figure 4A. From the comparison of the two controls, it is possible to verify that the addition of LUVs in PBS did not affect cells’ metabolic activity. Concentrations of LUVs lower than 5 mM were not cytotoxic for GBML42 cells. After determining free DOX efficacy in inducing GBM cell death (Table 2 and Appendix A) and evaluating the concentration of LUVs without toxic effects, we evaluated if DOX maintained its effectiveness after encapsulation in LUVs.

Figure 4B shows GBML42 metabolic activity in the presence of different concentrations of LUVs incorporating DOX. As expected, there were no significant differences between controls 1 and 2. Conversely, in the presence of the liposomes incorporating DOX, cells’ metabolic activity significantly decreased after 24 h of culture, being practically null for the remaining days. These results demonstrate the effectiveness of the liposome formulation as an anticancer therapy.

#### 2.3.3. Efficacy of the Hydrogel in Damaging GBML42 Cells

To determine the hydrogel cytotoxicity, GBML42 cells were seeded on top of the different formulations, namely, hydrogels of HA, HA with LUVs (HA+LUVs), and HA with LUVs with 4 µM of DOX (HA+LUVs+DOX). The concentration of DOX was selected according to the IC_50_ results. As observed in Figure 5A, GBML42 cells seeded on HA hydrogels containing or not 150 µM of LUVs presented an increase in metabolic activity over time, proving that these formulations are not cytotoxic. Conversely, the incorporation of LUVs with 4 µM of DOX on the HA hydrogels led to a significant decrease in GBM cells’ metabolic activity after the first 24 h of culture. Live/dead assays (Figure 5B) confirmed the results of GBML42 cells’ metabolic activity. Indeed, an increasing number of live cells in HA hydrogels containing or not LUVs was observed over time. However, the inclusion of DOX into the formulations led to a significant decrease in living cells. Moreover, at the last time points, it was not possible to observe any viable cells.

The metabolic activity and viability of GBML42 cells seeded on top of RGDS-HA hydrogels incorporating or not LUVs with or without DOX are presented in Figure 6A and Figure 6B, respectively. GBML42 cells’ metabolic activity decreased in RGDS-HA hydrogels, despite HA hydrogels not being cytotoxic. The live/dead assay confirmed these results. Therefore, the cytocompatibility of the peptide was evaluated.

GBML42 cells’ metabolic activity (Appendix A) and viability (Appendix A) in the presence of 500 ng/mL of RGDS were not statistically significantly different from the control. This RGDS concentration was selected based on the results obtained in the assays performed with 300 µL of functionalized HA.

As no cytotoxicity was observed, another experiment was performed. GBML42 cells were cultured for 24 h in the absence of FBS to assess if its presence affected the results obtained. Indeed, FBS in the medium can influence the results obtained due to the presence of a high amount of protein [45]. As observed in Figure 7A, cells presented a metabolic activity lower than that of control A but similar to that of control B. Moreover, after 7 days, cells were able to proliferate, reaching values like those of control A. Consequently, the decrease in their metabolic activity could be due to the absence of FBS in the medium instead of the presence of RGDS. Live/dead results (Figure 7B) confirmed the results of metabolic activity. In conclusion, RGDS in solution did not affect GBML42 cells’ metabolic activity and viability.

#### 2.3.4. Expression of MMP-2 by GBML42 Cells

Despite RGDS not being cytotoxic in solution, it was hypothesized that this peptide can be cytotoxic when released from the hydrogel to where cells are adhered. Indeed, the amide bond between RGDS and HA could be disrupted by proteolytic enzymes released from GBML42 cells, such as matrix metalloproteinases (MMPs). MMP-2 is an example of a protease involved in the progression of GBM, and its expression in normal brain tissue has not been observed [46].

The metabolic activity, proliferation, and expression of MMP-2 by GBML42 cells are shown in Figure 8. GBML42 cells’ metabolic activity increased until 7 days and remained similar after 10 days of culture (Figure 8A). Moreover, they were able to proliferate during the entire experimental period (Figure 8B).

MMP-2 expression by GBML42 cells (Figure 8C) increased until the end of the experiment, reaching a final value of 18.45 ± 0.46 ng/mL at day 10.

Considering the amount of MMP-2 expressed by the selected primary human cell line of GBM used for the in vitro tests after 72 h of culture, 4.57 ng/mL of this protease was incubated with the RGDS-HA hydrogel to demonstrate its ability to cleave the amide bond between RGDS and HA. The time points (24, 48, and 72 h) were selected considering the results of the metabolism and viability of cells when seeded on RGDS-HA hydrogels (Figure 6). As observed in Figure 9, the release of RGDS from the hydrogel was ±22% in the first 24 h and increased until the end of the experiment (72 h).

#### 2.3.5. Hydrogel Safety for Astrocytes

The toxicity of the developed formulation for healthy cells, namely, astrocytes (human immortalized astrocytes, hTERT/E6/E7 cell line), was also evaluated. In this assay, the concentration of DOX was reduced to 0.1 µM. This reduction was performed considering the IC_50_ values of DOX at 2, 3, and 7 days and its synergistic action with RGDS in killing GBML42 cells. To confirm the effectiveness of this formulation, GBM cells were seeded, as in the previous assays, on the hydrogel. As can be observed in Appendix A, the metabolic activity of GBML42 cells decreased after their culture on the hydrogel. These results were confirmed by the live/dead staining that showed a decrease in GBML42 cells’ viability (Appendix A).

The metabolic activity of hTERT/E6/E7 increased (Figure 10A), and the live/dead results also confirmed that astrocytes remained viable at all time points tested (Figure 10B). These results indicate that the developed strategy is not cytotoxic for astrocytes.

## 3. Discussion

Hydrogel formulations have a huge potential to be used as local delivery systems in GBM treatment. This study investigated whether a hydrogel of HA functionalized with RGDS and cross-linked with DPPC LUVs incorporating DOX can be used as an effective local treatment to eliminate GBM cells that remain after tumor resection. However, if DOX resistance is observed, other anticancer drugs (e.g., prazosin, cerulenin, or orlistat) can be encapsulated in the hydrogel. Indeed, the versatility of the developed formulation will enable, in the future, the production of personalized treatments based on the patient’s GBM fingerprint.

Liposomes were composed of a saturated phospholipid since their interaction with cancer cell membranes (exchange of lipids as well as adsorption, binding, internalization, or fusion of liposomes with the cell membrane) [47] could lead to a favorable decrease in their fluidity. Indeed, the membrane of GBM cells presents a high content of unsaturated fatty acids and, consequently, a high fluidity that may enhance tumor cell motility and invasion [48]. LUVs were able to encapsulate relevant concentrations of DOX considering the IC_50_ results (≈3.82 µM at 24 h). LUVs slightly increased in size after encapsulation of DOX (Table 1), and the PDI values of both formulations demonstrate that homogeneous suspensions were prepared [49]. DPPC is a zwitterionic phospholipid with a positively charged choline group attached to a negatively charged phosphate group [50]. The zeta potential of LUVs depends on the pH of the solution, and low pH values lead to positive zeta potential, whereas at high pH, the zeta potential turns negative. As this formulation of LUVs is in PBS (pH of 7.4), slightly negative values of zeta potential were obtained [51,52]. As they are negatively charged, the inclusion of LUVs in the matrix of HA will result in electrostatic repulsions with the anionic polymer, which can lead to a higher viscosity of the gels when compared to, for example, cationic liposomes [32]. Size homogeneity was also confirmed by STEM analyses (Appendix A) that demonstrated the round shapes of LUVs and LUVs+DOX. Furthermore, DPPC liposomes presented stability without significant changes in size and PDI for 10 days. The zeta potential presented some variation with time, possibly due to different interactions of liposomes with the salt (PBS buffer; Appendix A).

Regarding thermal stability (Appendix A), T_m_ indicates the maximum temperature at which liposomes undergo a transition from the ordered gel phase to the disordered liquid crystalline phase, increasing the permeability of the phospholipid membrane that can lead to the burst release of DOX. T_onset_ indicates the temperature at which this transition begins [53]. T_m_ for dehydrated DPPC, 63.85 ± 1.38 °C, is higher than that observed in the fully hydrated state, 42 °C, since, in the absence of water, the space between the phospholipid head groups decreases, giving rise to increased van der Waals interactions between the lipid hydrocarbon chains, and a greater T_m_ [54,55]. The encapsulation of DOX into the phospholipidic membrane decreased its T_m_ and T_onset_ (51.46 ± 0.35 °C and 49.11 ± 0.14 °C, respectively). This can be related to intermolecular interactions between the phospholipids and DOX, altering the properties and stability of the lipid bilayer [56].

HA hydrogels are macroscopic networks consisting of randomly interconnected polyanionic chains [57]. The conformation of HA molecules depends on their molecular weight, the pH of the environment, the nature of the ions, and the temperature. HA, at physiological ionic strength and pH, establishes electrostatic interactions with other substances, resulting in a certain spatial orientation of HA molecules [58]. Therefore, the functionalization of HA with RGDS modified the structural orientation of HA molecules and, consequently, the properties of the hydrogel. First, the successful functionalization of HA with RGDS was demonstrated by ^1^H-NMR (Appendix A) and ATR-FTIR (Appendix A) analyses. After functionalization, the hydrogels were mechanically and thermally characterized. The conformational characteristics of HA macromolecules define the hydrogel’s viscoelastic properties. The elastic properties of a hydrogel change with polymer concentration and molecular weight due to the entanglement of the chains. Rheology studies are, therefore, essential to evaluating the effect of derivatization on the HA network [58]. The binding of RGDS to HA by an amide bond altered the macroscopic network, leading to a hydrogel with much lower G′ and G″ moduli (Figure 1 and Appendix A). As mentioned before, this conformational alteration led to significant differences in the elastic and viscous properties of the hydrogel. The addition of liposomes to the network did not significantly affect the rheological properties of the resulting hydrogel. Moreover, the G′ values were always higher than the G″ values, indicating that the hydrogels display a predominantly elastic behavior rather than a viscous fluid-like behavior [59]. The increase in temperature from 25 to 37 °C slightly decreased G′. This difference was statistically significant in HA hydrogels (Figure 1 and Appendix A). The hydrodynamic volume of HA chains depends on the stiffness of the chains. HA chains gain flexibility when temperature increases, reducing the rigidity of the polysaccharide backbone and its hydrodynamic volume. This alteration in conformation causes a reduction in the elastic properties of HA [60,61]. Notably, the final formulation can be injectable and presents mechanical properties in the order of magnitude of the brain tissue [44].

The differential scanning calorimetry (DSC) thermograms of HA and RGDS-HA hydrogels also showed differences in both formulations (Appendix A and Appendix A). Both hydrogels presented an endothermic peak that is probably associated with the loss of remaining free water and water linked to the polymeric structure through hydrogen bonds [62,63]. Moreover, the DSC profiles show that after HA functionalization, the endothermic peak slightly shifted and the ∆H decreased. The relatively low ∆H obtained for RGDS-HA hydrogels may reflect lower water retention through the drying period [63]. This behavior can be related to the RGDS-HA mechanical properties (lower G′ values than HA hydrogels) since water absorbed into a softer hydrogel is desorbed to a greater extent [64].

Regarding DOX release (Figure 3), both formulations ensured a sustained release, but RGDS-HA hydrogels led to higher drug concentrations in the medium. The diffusion rate depends on the morphology of the network, the chemical composition of the hydrogel, the water content, the concentration of solubilized compounds, and the level of material swelling [58]. RGDS-HA hydrogels, when compared with HA hydrogels, have a lower G′ that can promote a higher diffusion rate of DOX.

After the characterization of the proposed formulations, their biological performance was investigated. First, the DOX IC_50_ was determined for the GBML42 cells. This GBM cell line has a very low value of IC_50_ for DOX, meaning that this drug is effective at low concentrations (Appendix A and Table 2). Afterward, the cytotoxicity of LUVs and LUVs incorporating DOX was assessed. Concentrations of empty LUVs lower than 5 mM were not cytotoxic for GBML42 cells (Figure 4A). Conversely, in the presence of LUVs incorporating DOX (Figure 4B), cells’ metabolic activity and viability decreased until a minimum, demonstrating the effectiveness of this formulation for the treatment of GBM.

GBML42 cells, when seeded on HA hydrogels containing or not 150 µM of LUVs, presented an increase in their metabolic activity (Figure 5A) and viability (Figure 5B) over time, demonstrating that these formulations are not cytotoxic. These results were expected since HA is one of the main components of the healthy brain and GBM’s TME [65]. The incorporation of LUVs with DOX into the HA hydrogel led to a significant decrease in cells’ metabolic activity (Figure 5A) and viability (Figure 5B) over time, demonstrating, again, the efficiency of DOX.

GBML42 cells seeded on RGDS-functionalized HA hydrogels incorporating or not LUVs led to a decrease in cells’ metabolic activity (Figure 6A) and viability (Figure 6B). Therefore, the cytocompatibility of the peptide in solution was evaluated (Appendix A). As expected, the free peptide did not present cytotoxicity. Nonetheless, the presence of FBS in the medium could influence the results obtained due to the presence of a high number of proteins. Indeed, in the absence of FBS, RGDS was shown to be cytotoxic for melanoma cells [45]. Thus, cells were starved for 24 h and then treated with RGDS for 48 h. Even though the metabolic activity was inferior when compared with cells without treatment in the presence or absence of RGDS, after 7 days of culture, cells were able to proliferate, reaching the values of the control performed with FBS (Figure 7A). These results were confirmed by live/dead images (Figure 7B). This demonstrates that the decreased cells’ metabolic activity was caused by the absence of FBS and not by the presence of RGDS. Consequently, this peptide is not cytotoxic for GBM cells in solution.

Even though RGDS is not cytotoxic in solution, it was hypothesized that this peptide can be cytotoxic if released from the hydrogel where cells are adhered to, causing apoptosis of invasive GBM cells. Indeed, the covalent bond between the peptide and HA can be cleaved by proteolytic enzymes released by GBML42 cells [66,67]. GBM cells can secrete several proteolytic enzymes for the destruction of the ECM, such as MMPs, whose expression is not observed in the normal brain [46,68]. Particularly, MMP-2 and MMP-9 are MMPs overexpressed in GBM that are involved in tumor progression [68]. Consequently, the expression of MMP-2 by GBML42 cells was investigated. It was observed that MMP-2 expression increased with cells’ proliferation (Figure 8). Moreover, MMP-2 was able to cleave the RGDS-HA chemical bond, and the concentration of the peptide released from the hydrogel increased with time (Figure 9). Consequently, the hydrogel cytotoxicity can be related to RGDS internalization, where the cells that have adhered to [45]. RGDS can alter the apoptotic intracellular signaling pathways in cancer cells, which can be beneficial for treatment outcomes. Indeed, this peptide can accumulate in cells’ cytosolic compartments and activate the apoptotic cascade [45,69]. For instance, RGDS peptide can be internalized by melanoma cells interacting with pro-caspase 3, 8, and 9 (proteases that regulate apoptosis) and survivin, causing apoptosis [45]. Further investigation is warranted in future studies regarding the effect of the RGDS peptide in GBM cells, namely, the interaction of RGDS with GBML42 cell caspases.

Despite having good efficacy in inducing cancer cell death, the developed formulation was not cytotoxic for astrocytes in a co-culture system. Indeed, this co-culture tried to mimic what occurs in vivo after GBM resection. The metabolic activity of the human immortalized astrocyte hTERT/E6/E7 cell line in the presence of the several formulations was similar to the control at all time points tested (Figure 10A) and remained viable for 10 days (Figure 10B). In conclusion, the system developed was not cytotoxic for healthy cells. Therefore, this formulation presents potential as an efficient and safe alternative treatment that should be tested in vivo. Future assays will include the assessment of the hydrogel efficacy in different GBM cells and neural cells, as well as in more complex 3D cell models, before being tested in vivo.

## 4. Materials and Methods

### 4.1. Materials

Sodium hyaluronate was purchased from Lifecore Biomedical (molecular weight-MW 1.64 MDa, Chaska, MN, USA). DPPC, support filters, and polycarbonate membranes (100 nm in pore size) were purchased from Avanti Polar Lipids (Alabaster, AL, USA) and the LabAssay Phospholipid kit from FUJIFILM Wako Pure Chemical Corporation (Tokyo, Japan). DOX hydrochloride was purchased from MedChemExpress (Monmouth Junction, NJ, USA). Human matrix metalloproteinase-2 (MMP-2) was purchased from Eurogentec (Seraing, Belgium), and the human MMP-2 DuoSet ELISA was purchased from Research and Diagnostic Systems (Minneapolis, MN, USA). Formic acid was purchased from PanReac Applichem (Chicago, IL, USA), and material dialysis devices (molecular weight cutoff (MWCO) of 3.5–5 kDa) and dialysis membranes (MWCO 5 kDa) were purchased from SpectrumLabs (Piraeus, Greece). PD-10 desalting columns were purchased from GE Healthcare (Chicago, IL, USA). 1-hydroxybenzotriazole monohydrate (HOBt) hydrate, deuterium oxide (D_2_O), diethyl ether, phosphoric acid (85 wt. % in H_2_O, 99.99% trace metals basis), Dulbecco’s Modified Eagle’s Medium (DMEM) with low glucose, ethanol absolute, N-(3-dimethylaminopropyl)-N′-ethylcarbodiimide (EDC), N-hydroxysuccinimide (NHS), N,N-diisopropylethylamine (DIPEA), 2-(N-morpholino)ethanesulfonic acid (MES) hydrate ≥99.5% (titration), phosphate buffered saline (PBS) tablets, and trifluoracetic acid (TFA) were purchased from Sigma-Aldrich (Burlington, MA, USA). Acetonitrile (CHROMASOLV, for HPLC, gradient grade, ≥99.9%) and dimethylformamide (DMF) were purchased from Fisher Scientific (Waltham, MA, USA). Roswell Park Memorial Institute (RPMI) 1640 Medium, fetal bovine serum (FBS), antibiotic/antimycotic, Quant-ITTM PicoGreen dsDNA Assay Kit, Live/Dead cell double staining kit, microBCA kit, and triisopropylsilane (TIS) were purchased from ThermoFisher (Waltham, MA, USA). Hexafluorophosphate benzotriazole tetramethyl uronium (HBTU) and Fmoc-amino acids (Fmoc-Gly-OH, Fmoc-Asp(OtBu)-OH, Fmoc-Ser(tBu)-OH, and Fmoc-Arg(Pbf)-Wang resin) were purchased from Novabiochem (Washington, DC, USA). The 3-(4,5-dimethylthiazol-2-yl)-5-(3-carboxymethoxyphenyl)-2-(4-sulfophenyl)-2Htetrazolium (MTS) and the AlamarBlue were purchased from Promega (Madison, WI, USA) Bio-rad (Amadora, Portugal), respectively. Aluminum pans and transmission electron microscopy (TEM) grids (Carbon Type-B 400M Cu, Ted Pella) were purchased from STECinstruments (Austin, TX, USA) and IESMAT, respectively. Inserts (0.4 µm of pore size) were purchased from Greiner Bio-One (Kremsmünster, Austria).

### 4.2. Production of DPPC Liposomes with and without DOX

LUVs of DPPC with and without DOX hydrochloride were prepared by the thin film hydration method [70], followed by extrusion using a mini extruder (Avanti Polar Lipids, Alabaster, AL, USA). LUVs encapsulating DOX were hydrated with a solution of this anticancer drug in PBS. To produce LUVs, multilamellar vesicles (MLVs) were extruded 43 times through polycarbonate membranes of 100 nm pore size at 43–45 °C. Non-encapsulated DOX was removed through gel filtration chromatography using a PD-10 Desalting Column. The final concentration of DPPC was determined with the LabAssay Phospholipid kit, according to manufacturer instructions.

### 4.3. Characterization of LUVs

#### 4.3.1. Determination of DOX Concentration in LUVs

The concentration of the encapsulated DOX was determined by HPLC (Alliance 2695, Waters, Milford, MA, USA). A column sb-c18 (Zorbax, Agilent, Santa Clara, CA, USA) was used as the stationary phase. The isocratic mobile phase consisted of a mixture of 70% ultra-pure water (with pH adjusted to 3 with phosphoric acid) with 30% acetonitrile. The flow rate was 1 mL/min, and the injection volume was 10 µL with a run time of 10 min. The column temperature was kept at 30 °C, and the detector signal was monitored at a wavelength of 254 nm. Samples were diluted with ethanol and phosphoric acid (1:500), and standard samples of DOX in PBS, ethanol, and phosphoric acid were prepared by diluting the stock solution of 1 mM of DOX in PBS.

#### 4.3.2. Determination of Size, PDI, Zeta Potential, and Stability

The size, PDI, and zeta potential of empty LUVs and LUVs incorporating DOX were determined using a Zetasizer Nano ZS (Malvern Instruments, Malvern, UK) at 37 °C. The stability and integrity of the LUVs were evaluated by measuring, for 10 days, their size, PDI, and zeta potential. The samples were analyzed in disposable polystyrene cuvettes at a concentration of 500 µM of LUVs in PBS. A dip-cell (Malvern Instruments, UK) was inserted in the cuvette to measure the LUVs’ zeta potential. Each measurement was performed at least three times.

#### 4.3.3. Thermal Characterization

The thermal properties of LUVs and LUVs incorporating DOX were studied with the DSC Q100 (TA Instruments, New Castle, DE, USA). To determine T_m_ and T_onset_, samples were frozen and freeze-dried (LyoQuest-80 Plus ECO, Telstar, Barcelona, Spain). Afterward, the samples were transferred to aluminum pans (4–5 mg), which were subsequently sealed and analyzed. The samples were stabilized at 10 °C for 5–10 min and scanned from 10 to 70 °C at 2 °C/min.

#### 4.3.4. LUV Morphology

The morphology of LUVs incorporating or not DOX was studied with STEM using a High-Resolution Field Emission Scanning Electron Microscope with Focused Ion Beam (Auriga Compact, Zeiss, Jena, Germany). The samples were diluted to a concentration of 10 µM. A drop of each sample was put on top of TEM grids, left to dry overnight at room temperature, and analyzed the following day.

### 4.4. RGDS Synthesis

RGDS was synthesized using a solid phase approach following the Fmoc strategy in an automated peptide synthesizer (CS Bio, Menlo Park, CA, USA). Wang resin was used as a solid support, and the conjugation of the first amino acid was achieved using DIPEA/HOBt as coupling reagents dissolved in DMF. The full peptide sequence was obtained by sequential conjugation of the respective Fmoc-protected amino acids. The full peptide was cleaved from the resin with a cleavage cocktail of TFA:TIS:H_2_O (0.95:0.25:0.25, v/v) for 3 h and 30 min. The solvent was evaporated, and the crude product was collected after precipitation in cold diethyl ether. Then, the solid was filtered and washed again with cold diethyl ether. The peptide was characterized by ultra-performance liquid chromatography mass spectrometry (UPLC-MS; Acquity UPLC H-Class, Waters, Milford, MA, USA, connected to a mass spectrometer, Waters Quattro Micro API, Milford, MA, USA). When needed, the peptides were purified by preparative UPLC-MS, and the TFA salt was exchanged using an acid removal filter (VariPure™ IPE Ion Pair Extraction, Stockport, UK).

### 4.5. Functionalization of HA with RGDS

HA was functionalized with RGDS through carbodiimide chemistry. For the activation of HA carboxylic acid groups, EDC and NHS in MES buffer, at pH 6.5, with a molar concentration ratio of 1:2, were added. The molar ratio of HA to EDC was 1:25. After 15 min of stirring, RGDS was added to this solution. After overnight incubation at 4 °C, dialysis was made for 7 days with NaCl and, then, with H_2_O to remove unreacted components. Afterward, the functionalized HA was freeze-dried and stored at −20 °C. HA functionalization was confirmed by NMR spectroscopy (Avance II, Bruker, Billerica, MA, USA) and ATR-FTIR spectroscopy (IR Prestige 21, Shimadzu, Kyoto, Japan).

For NMR analyses, HA, RGDS, and RGDS-functionalized HA were dissolved in D_2_O. NMR spectra were obtained at a resonance frequency of 400 MHz for ^1^H and 100.6 MHz for ^13^C, using the D_2_O solvent peak as an internal reference at 70 °C. D_2_O was used as a reference for chemical shifts, which are reported in ppm.

In ATR-FTIR, a small amount of RGDS-functionalized HA, HA, and RGDS was analyzed with 30 scans between 400 and 4000 cm^−1^. The full spectrum was subjected to a baseline correction algorithm performed using the IRsolution software version 1.04.

### 4.6. Preparation of Hydrogels

Hydrogels of HA were prepared at different concentrations, namely, 1, 2.5, and 5% (w/v) by its dissolution in PBS. Afterward, 5% (w/v) HA and RGDS-functionalized HA hydrogels with and without 150 µM of LUVs were also prepared by dissolving the polymer in the liposome suspension.

### 4.7. Characterization of HA and RGDS-Functionalized HA Hydrogels

#### 4.7.1. Mechanical Characterization

The mechanical characterization of the hydrogels was performed on the rheometer Kinexus Prot (Malvern Instruments, UK) with a plate geometry of 20 mm in diameter. Amplitude sweep strain-controlled tests at a frequency of 1 Hz were performed to determine the linear viscoelastic region (LVR). Subsequent frequency sweep tests were performed between 0.1 and 100 Hz using a constant strain within the LVR of 1% to determine the G′ and G″ moduli. The tests were performed at 25 °C and 37 °C with three independent measurements per sample.

#### 4.7.2. Thermal Characterization

HA and RGDS-functionalized HA hydrogels with and without 150 µM of LUVs were frozen and freeze-dried (LyoQuest-80 Plus ECO, Telstar, Barcelona, Spain) before being analyzed in the DSC Q100 (TA Instruments). The samples were transferred into aluminum pans (4–5 mg) and scanned from 0 to 200 °C at 10 °C/min.

#### 4.7.3. Cleavage of the RGDS-HA Bond by MMP-2

RGDS-functionalized HA hydrogels at 5% (w/v) were inserted into dialysis devices (MWCO 3.5–5 kDa) with 4.57 ng/mL of human MMP-2. A control condition of RGDS-HA hydrogels without MMP-2 was performed. Then, these devices were dipped in PBS at 37 °C. Samples of 1 mL were collected at intervals of 24 h for 3 days. The concentration of RGDS released from the hydrogel by MMP-2 was determined with a microBCA kit, according to the manufacturer’s specifications. Then, the RGDS amount released in the presence of the enzyme was subtracted from the quantity released in the control condition.

#### 4.7.4. Determination of DOX Release by the Hydrogel

The 5% (w/v) HA and RGDS-functionalized HA hydrogels with 1 mM of LUVs incorporating DOX at an initial concentration of 84.54 ± 15.86 μM were placed in dialysis devices (MWCO of 3.5–5 kDa) that were further dipped in PBS at 37 °C. At determined time points (intervals of 15 min until 1 h, intervals of 1 h in the first 7 h, and 24 h intervals for the remaining 10 days), 200 µL of PBS was collected. The UPLC-MS referred to in Section 4.4 was used to determine the concentration of DOX released by the hydrogel.

The MS spectrum was acquired under the following parameters: capillary voltage of 3.25 kV, cone 20 V, extractor 2 V, RF lens 0.5 V, source temperature of 125 °C, and a desolvation temperature of 350 °C. The cone gas and desolvation gas flows used were 50 L/h and 700 L/h, respectively. The function used was MRM of 2 channels with two transitions, namely, 544.00 > 130.00 and 544.00 > 397.00, with a collision energy of 13 eV. In the UPLC system, an Atlantis T3 column (Waters, USA) was used as the stationary phase. The isocratic mobile phase consisted of a mixture of 50% (v/v) water with formic acid 0.1% and 50% (v/v) acetonitrile with formic acid 0.1%. The flow rate of the mobile phase was 0.5 mL/min. The detector signal was monitored at a wavelength of 254 nm. The injection volume was 40 µL, with a run time of 6.5 min.

### 4.8. In Vitro Assays to Assess the Safety and Efficacy of the Hydrogel

#### 4.8.1. Cell Culture

The GBM patient-derived primary culture GBML42 and the human immortalized astrocyte hTERT/E6/E7 cell line were used in the in vitro experiments. The GBM patient-derived primary culture GBML42 was established by Bruno M. Costa’ laboratory [71]. It was established from freshly collected GBM tissue recovered from a patient undergoing surgery at the Hospital de Braga. The human immortalized astrocytes (hTERT/E6/E7) were established by retrovirally infecting isolated human astrocytes with constructs encoding the human telomerase catalytic component (hTERT), and the human papillomavirus 16 E6/E7 (to inactivate both p53 and pRb) and were kindly provided by Dr. Russell O. Pieper (University of California, San Francisco) to Dr. Bruno M. Costa.

The GBML42 and hTERT/E6/E7 cells were cultured in RPMI 1640 and DMEM media (except in the co-culture assay where RPMI was used for both cells), respectively, supplemented with 10% FBS and 1% antibiotic/antimycotic. The medium was replaced every 2 to 3 days, and cells were passed when an approximate 80–90% confluence was reached. Cells were incubated at 37 °C in a humidified atmosphere containing 5% CO_2_.

#### 4.8.2. Determination of DOX IC_50_ in GBML42 Cells

GBML42 cells (5 × 10^4^ cells/well) were seeded in 48-well plates and incubated for 24 h. Afterward, they were treated with different concentrations of DOX (0.001, 0.005, 0.01, 0.05, 0.1, 0.5, 1, 5, 10, 50, and 100 µM). Cells’ metabolic activity was assessed after 1, 2, 3, and 7 days. At each time point, cells were incubated with medium containing MTS, 1:5 (v/v), for 3 h at 37 °C and 5% CO_2_. The absorbance was measured at 490 nm in a microplate reader (Synergy HT, Bio-Tek Instruments, Winooski, VT, USA). IC_50_ [72] was determined using GraphPad Prism (software version 6.00).

#### 4.8.3. Metabolic Activity of GBML42 Cells in the Presence of LUVs Encapsulating or Not DOX

GBML42 cells were cultured at an initial density of 5 × 10^4^ cells/well in 48-well plates and incubated at 37 °C and 5% CO_2_. After 24 h, the medium was replaced by medium with different concentrations of LUVs (10, 7.5, 5, 2.5, and 1 mM) and LUVS (1, 0.5, 0.25, 0.015, and 0.05 mM) incorporating DOX. Then, the plates were incubated for 1, 3, 7, and 10 days. Two controls were performed for each assay. In control 1, cells were treated only with RPMI 1640 medium, and in control 2, cells were treated with medium diluted with the maximum amount of PBS present in the condition with the highest concentration of LUVs. Cells’ metabolic activity was assessed using the MTS assay, as previously described.

#### 4.8.4. Efficacy of Hydrogels in Damaging GBML42 Cells

HA and RGDS-functionalized HA at 5% (w/v), incorporating or not LUVs (150 µM) containing or not DOX (4 µM), were produced and transferred to 48-well plates (300 µL per well). Afterward, GBML42 cells (5 × 10^4^ cells/well) were seeded on the top of the hydrogels and incubated at 37 °C. Cells’ metabolism and viability were assessed with MTS and live/dead assays, respectively, at the end of each time point (1, 3, 7, and 10 days). The MTS assay was performed as previously described. For the live/dead assay, a calcein AM 1:1200 (v/v) and PI 1:500 (v/v) solution in PBS was added to each well (50 µL). After 30 min of incubation at 37 °C protected from light, the samples were analyzed using a fluorescence inverted microscope (Axio Imager Z1, Zeiss, Germany).

#### 4.8.5. Cytotoxicity of RGDS

GBML42 cells were seeded at an initial density of 5 × 10^4^ cells/well and incubated at 37 °C and 5% CO_2_. After 24 h, the medium was replaced with a medium containing 500 ng/mL of RGDS. At each time point, 1, 3, 7, and 10 days, the metabolism and viability of GBML42 cells were determined with MTS and a live/dead assay, respectively, as described before.

To avoid FBS interference, cells were also seeded at an initial density of 5 × 10^3^ cells/well, 8 × 10^3^ cells/well, and 2.5 × 10^4^ cells/well in 48-well plates and incubated for 24 h. These densities were selected to study the effect of RGDS on higher cell densities compared with 5 × 10^4^ cells/well. After 24 h, the medium was replaced with a medium without FBS, and after 24 h replaced with a complete medium containing 500 ng/mL of RGDS. The metabolism of cells was assessed with MTS, and the cells’ viability was evaluated using a live/dead cell kit, as described before, after 2 and 7 days of culture.

Two controls were performed, namely, control A, which corresponds to cells cultured only with medium, and control B, where cells were treated with medium without FBS for 24 h and then, replaced with fresh medium. 

#### 4.8.6. MMP-2 Production by GBML42 Cells

GBML42 cells with a density of 5 × 10^4^ cells/well were seeded in a 48-well plate and incubated at 37 °C. At the end of each time point (1, 3, 7, and 10 days), the medium was collected and stored at −80 °C until further analysis. The amount of MMP-2 expressed by GBML42 cells was measured with a human MMP-2 DuoSet ELISA. The assay was performed according to the manufacturer’s recommendation. GBML42 cells’ metabolism and proliferation were assessed with AlamarBlue and DNA assays, respectively. For the AlamarBlue assay, the culture medium was removed from each sample and replaced with 300 μL of the 1:10 (v/v) solution of AlamarBlue in cell medium. Cells were then incubated for 3 h at 37 °C and protected from light. Then, the absorbance was measured in a microplate reader (Synergy HT, Bio-Tek Instruments, USA) at wavelengths of 570 and 600 nm. Afterward, each well was washed with 500 μL of PBS. Then, 1 mL of water was added to each well, and, after 30 min, the plate was kept at −80 °C until the determination of DNA concentration. In the DNA assay, the manufacturer’s recommended procedure was followed. The fluorescence of the samples was measured at excitation and emission wavelengths of 485 and 528 nm, respectively.

#### 4.8.7. Hydrogel Efficacy and Safety in a Co-Culture of GBML42 Cells and Astrocytes

RGDS-functionalized HA hydrogels at 5% (w/v) with and without LUVs encapsulating or not 0.1µM DOX were transferred to 24-well plates (200 µL per well). GBML42 cells (8 × 10^4^cells/well) were seeded on the top of the hydrogels. Afterward, the human immortalized astrocyte hTERT/E6/E7 cell line was seeded (1.5 × 10^5^ cells/well) in 24-well plate inserts with 0.4 µm of pore size. Then, they were placed in each well containing the hydrogels seeded with GBML42 cells on top. After cell culture (Section 4.8.1) for 1, 3, 7, and 10 days, the inserts were removed, and the metabolism and viability of the GBM cells and astrocytes were assessed separately using MTS and live/dead assays, respectively, as described before.

### 4.9. Statistical Analysis

Results are shown as arithmetic means ± standard deviation (SD) of at least three independent measurements. The software used for the statistical analyses was GraphPad Prism software version 6.00 (GraphPad Software, La Jolla, CA, USA) using t-tests, one-way analyses of variance (ANOVA), or a two-way ANOVA. Tests were selected according to the number and size of the samples and groups. Statistically significant differences were considered at *p* < 0.05 (the minimum of the 95% confidence interval).

## 5. Conclusions

In this study, we demonstrated that the fibronectin inhibitor RGDS can be positively used in the development of novel and effective therapies for GBM. Indeed, RGDS-functionalized HA hydrogels presented cytotoxicity even without DOX incorporation. As proteolytic enzymes such as MMP-2 can disrupt the peptide–HA bond, the possible internalization of free RGDS and consequent GBM cell apoptosis should be analyzed in the future. Importantly, RGDS-functionalized HA hydrogels incorporating liposomes with DOX efficiently damaged GBM cells without affecting the metabolism and viability of astrocytes, proving their safety. Consequently, the RGDS peptide could inhibit pro-tumorigenic signals associated with fibronectin and increase the cytotoxicity of new therapeutic formulas for GBM.

## Figures and Tables

**Figure 1 ijms-25-04910-f001:**
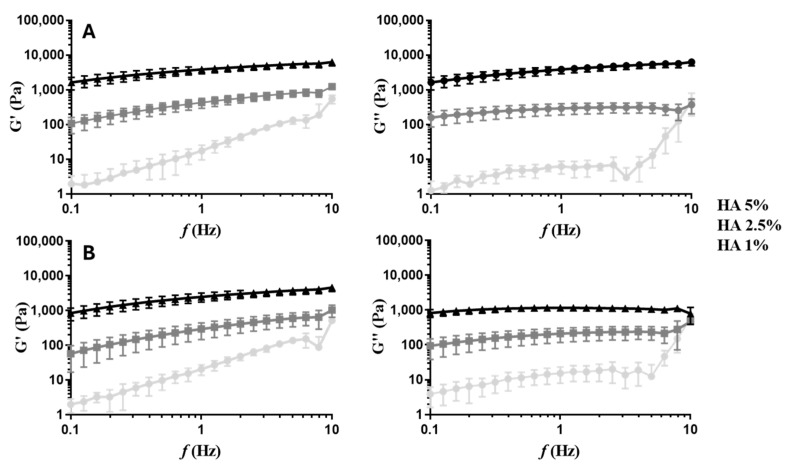
Elastic (G′) and viscous (G″) moduli of 1%, 2.5%, and 5% (w/v) HA at 25 °C (**A**) and 37 °C (**B**) with 1% strain.

**Figure 2 ijms-25-04910-f002:**
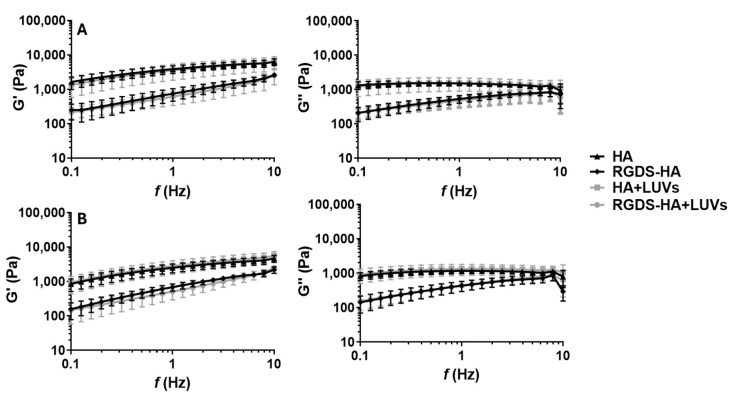
Elastic (G′) and viscous (G″) moduli of HA hydrogels before (HA) and after functionalization with RGDS (RGDS-HA) and without or with LUVs in their composition (HA+LUVs and RGDS-HA+LUVs) at 25 °C (**A**) and 37 °C (**B**) with 1% strain.

**Figure 3 ijms-25-04910-f003:**
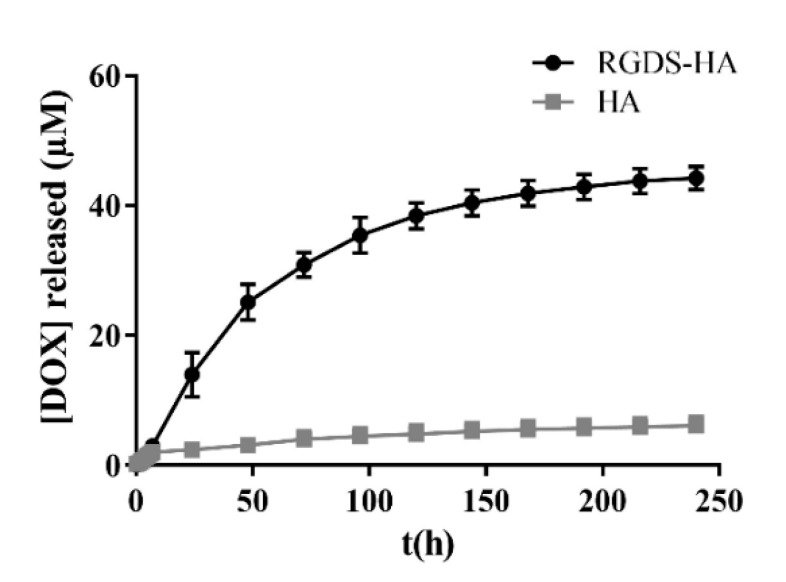
DOX release by HA and RGDS-HA hydrogels with 1 mM of LUVs.

**Figure 4 ijms-25-04910-f004:**
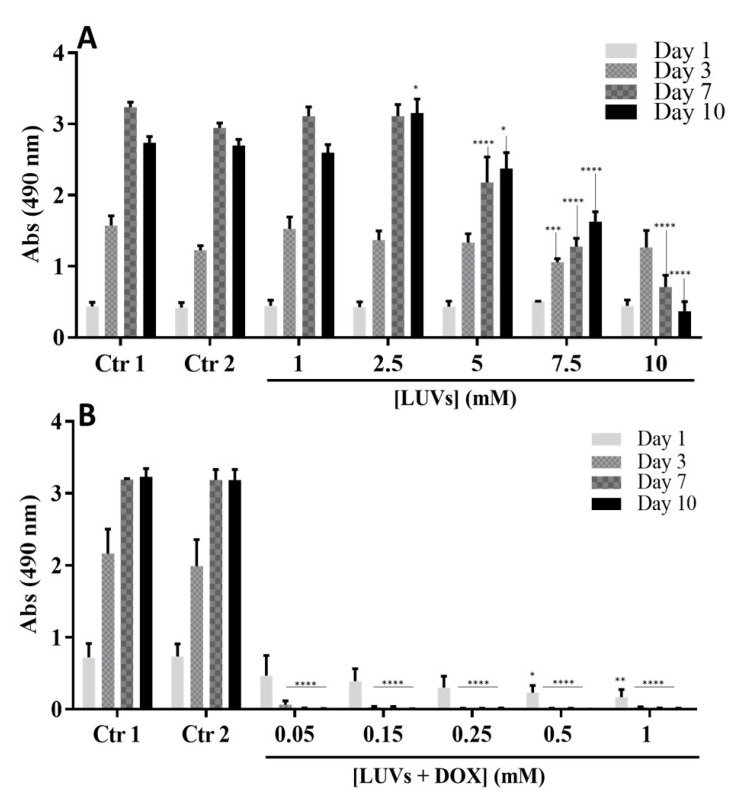
Metabolic activity of GBML42 cells incubated with different concentrations of LUVs ((**A**): 1, 2.5, 5, 7.5, and 10 mM and (**B**): 0.05, 0.15, 0.25, 0.5, and 1 mM) without (**A**) or incorporating DOX (**B**). Control (Ctr) 1 is obtained from the cell culture only with medium (without treatment), and Ctr 2 is when cells are exposed to a mixture of medium and PBS volume added in the 10 (**A**) or 1 (**B**) mM LUV conditions. The symbol (*) denotes significant differences versus Ctrs 1 and 2: **** *p* < 0.0001, *** *p* < 0.001; ** *p* < 0.01; * *p* < 0.05.

**Figure 5 ijms-25-04910-f005:**
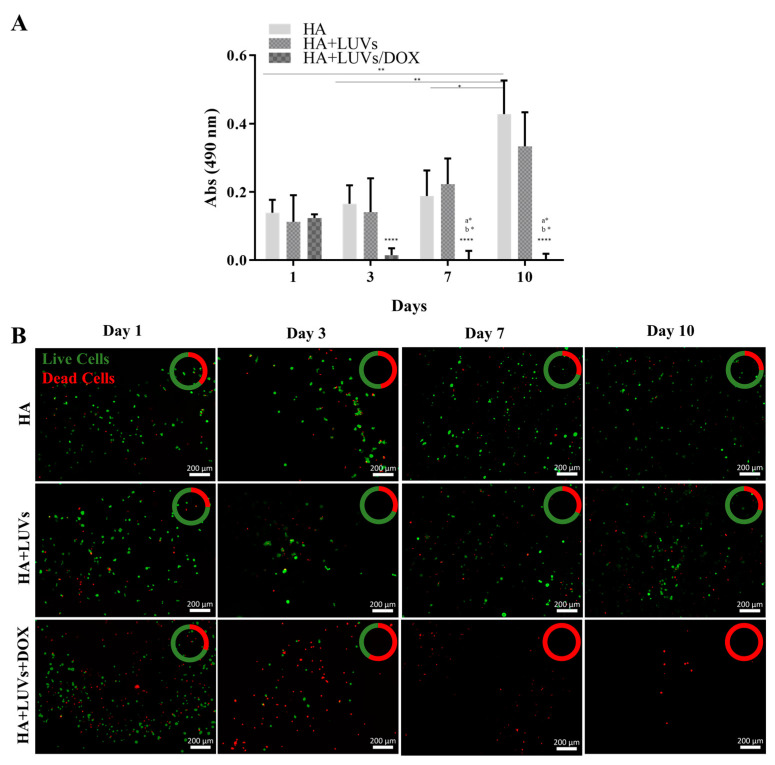
Metabolic activity (**A**) and fluorescence microscopy images at 5× magnification (**B**) of live/dead staining with Calcein AM/propidium iodine (PI) (green: live cell; red: dead cell) of GBML42 cells seeded on HA hydrogels incorporating or not 150 µM of empty LUVs (HA+LUVs) or encapsulating 4 µM of DOX (HA+LUVs+DOX). In (**A**), the symbol (*) denotes significant differences in the different days for each condition: (a) significant differences between HA and HA+LUVs+DOX, and (b) significant differences between HA+LUVs and HA+LUVs+DOX: **** *p* < 0.0001, ** *p* < 0.01; * *p* < 0.05. In (**B**), the amount of live cells (green) is present in comparison to dead cells (red). Scale bar of 200 µm.

**Figure 6 ijms-25-04910-f006:**
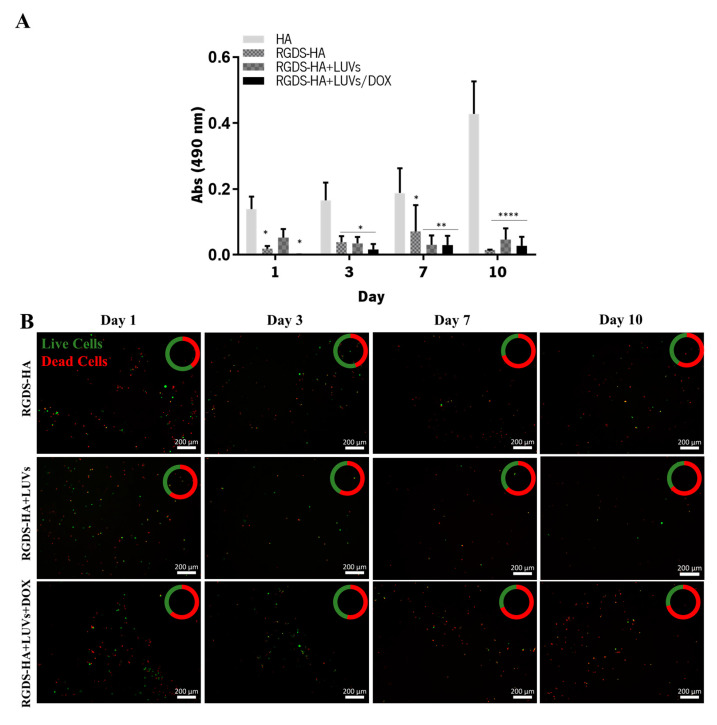
Metabolic activity (**A**) and fluorescence microscopy images at 5× magnification (**B**) of live/dead staining with Calcein AM/PI (green: live cell; red: dead cell) of GBML42 seeded on HA (fluorescence microscope images in Figure 5B), RGDS-functionalized HA (RGDS-HA) hydrogels incorporating or not 150 µM of empty LUVs (RGDS-HA+LUVs) or encapsulating 4 µM of DOX (RGDS-HA+LUVs+DOX). In (**A**), the symbol (*) denotes significant differences versus HA: **** *p* < 0.0001, ** *p* < 0.01; * *p* < 0.05. In (**B**), the amount of live cells (green) is present in comparison to dead cells (red). Scale bar of 200 µm.

**Figure 7 ijms-25-04910-f007:**
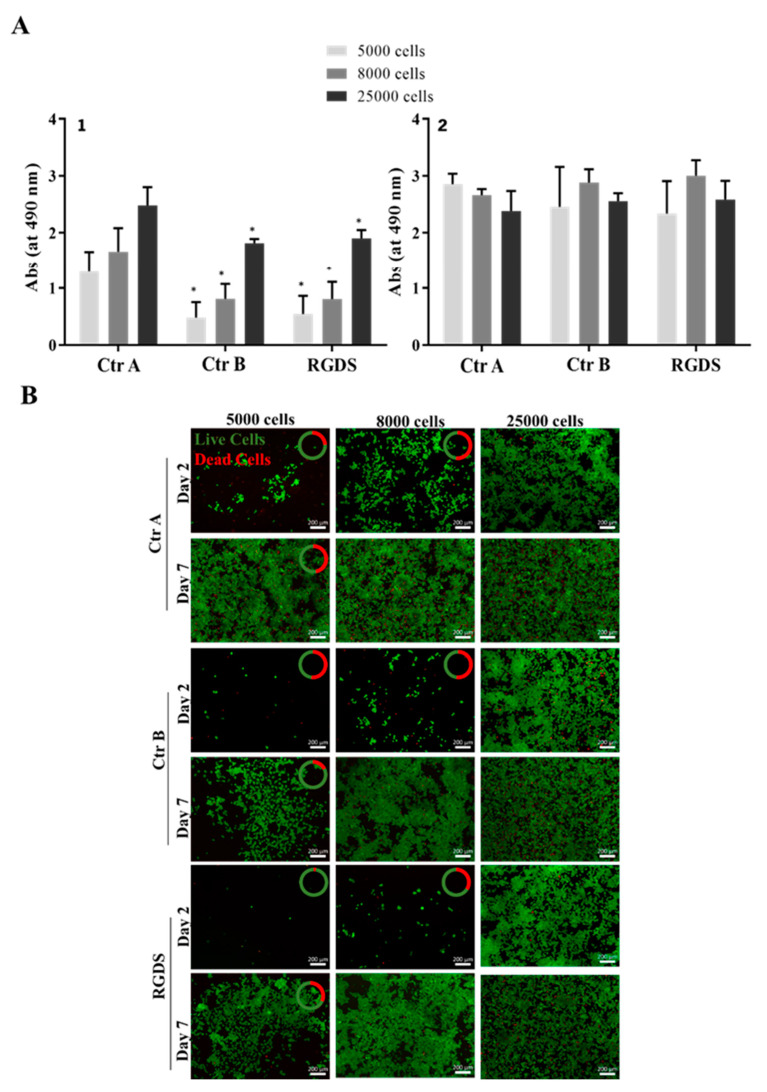
Metabolic activity (**A**) and fluorescence microscopy images at 5× magnification (**B**) of live/dead staining with Calcein AM/PI (green: live cell; red: dead cell) of GBML42 cells (5 × 10^3^, 8 × 10^3^, and 2.5 × 10^4^ cells/well on 48-well plates) in the presence of RGDS after 2 (**A1**) and 7 (**A2**) days of culture. Ctr A is of cells cultured only with medium, and Ctrl B is of cells treated with medium without FBS for 24 h and then replaced with fresh medium. In (**B**), the amount of live cells (green) is present in comparison to dead cells (red). The symbol (*) denotes significant differences versus Ctr 1: * *p* < 0.05. Scale bar of 200 µm.

**Figure 8 ijms-25-04910-f008:**
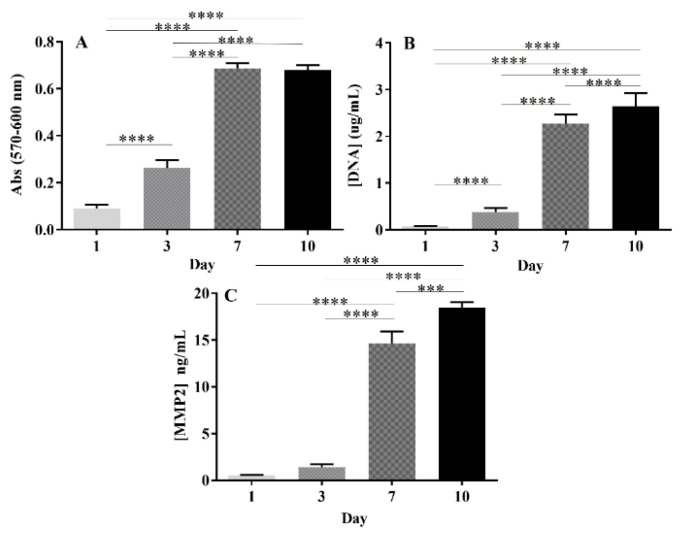
GBML42 cell metabolic activity (**A**), proliferation (**B**), and MMP-2 production (**C**). The symbol (*) denotes significant differences versus the different days: **** *p* < 0.0001; *** *p* < 0.001.

**Figure 9 ijms-25-04910-f009:**
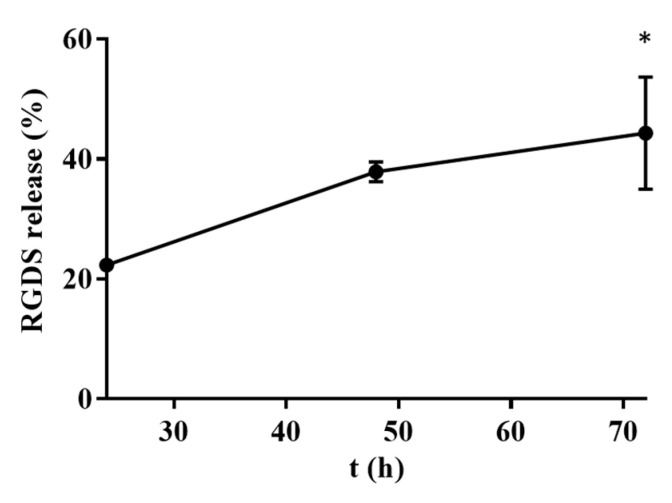
RGDS release (%) from the functionalized hydrogel in the presence of MMP-2 for 24, 48, and 72 h. The symbol (*) denotes significant differences versus the 24 h time point: * *p* < 0.05.

**Figure 10 ijms-25-04910-f010:**
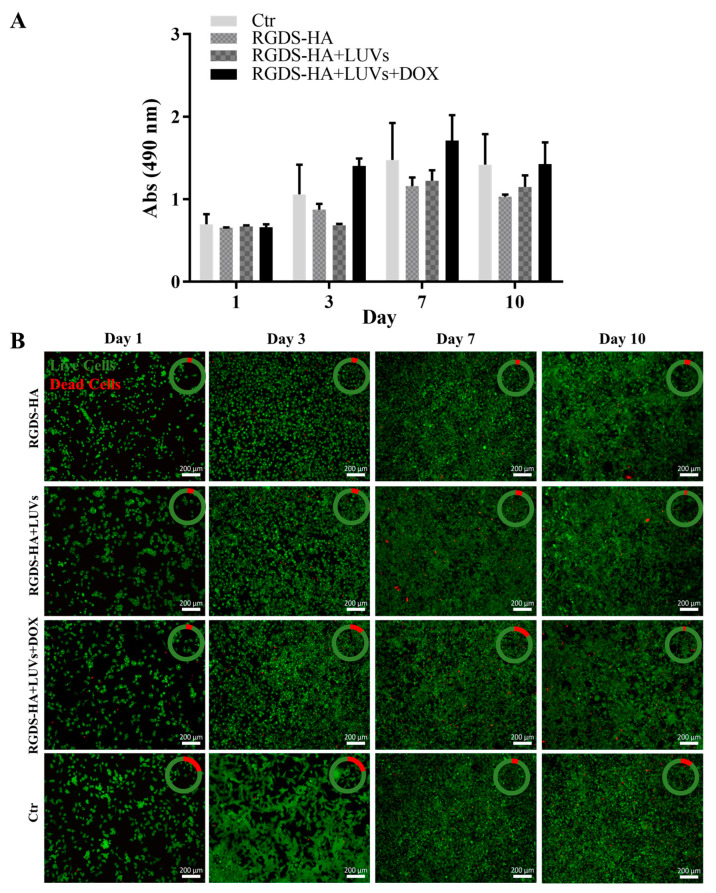
Metabolic activity (**A**) and fluorescence microscopy images at 5× magnification (**B**) of live/dead staining with Calcein AM/PI (green: live cell; red: dead cell) of human immortalized astrocytes hTERT/E6/E7 cell line cultured on 48-well culture plates or seeded in co-culture with GBML42 cells cultured on RGDS-functionalized HA (RGDS-HA) without or with 150 µM LUVs (RGDS-HA+LUVs) not or encapsulating 0.1 µM DOX (RGDS-HA+LUVs+DOX). In (**B**), the amount of live cells (green) is present in comparison to dead cells (red). Scale bar of 200 µm.

**Table 1 ijms-25-04910-t001:** Size, PDI, and zeta potential of empty LUVs and incorporating DOX (LUVs+DOX).

Sample	Size (nm)	PDI	Zeta Potential (mV)
LUVs	114.7 ± 2.3	0.076 ± 0.023	−1.38 ± 0.99
LUVs+DOX	121.7 ± 4.7	0.164 ± 0.041	−2.43 ± 1.33

**Table 2 ijms-25-04910-t002:** IC_50_ of DOX for GBML42 cells at different time points (1, 2, 3, and 7 days).

Day	IC_50_ (µM) ^1^
1	3.822 (3.089–4.730)
2	0.160 (0.136–0.189)
3	0.128 (0.114–0.144)
7	0.133 (0.116–0.152)

^1^ IC_50_ values indicate the DOX concentration (μM; mean (95% CI)).

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
