# Peer review of "The Potential of the Fibronectin Inhibitor Arg-Gly-Asp-Ser in the Development of Therapies for Glioblastoma"

_ijms, 2024, doi:10.3390/ijms25094910_

Round 1

Reviewer 1 Report

Comments and Suggestions for Authors

Castro-Ribeiro et al study the cytotoxic effect of RGDS in GBM cells. Before the manuscript is suggested for publication in the journal, some concerns need to be addressed.

It is needed to determine whether RGDS-HA can induce apoptosis in GBM under conditions of MMP2 knockdown or the use of MMP2 inhibitors."

All abbreviations should be written out in full on first use. For example, PDI and DPPC.

When referring to concentration, "lower" is the more appropriate term than "smaller."

Comments on the Quality of English Language

 Extensive editing of English language required.

Author Response

Author's Reply to the Reviewer 1 Report:

We kindly acknowledge and appreciate the comments and suggestions of Reviewer 1 that enabled us to improve and clarify our manuscript. The updated version of the manuscript ijms-2951342_Revised.doc includes the modifications based on the comments and suggestions proposed by the Reviewer, which were all considered and discussed in detail. In this updated version of the manuscript, the modifications proposed are marked in a different color.

Reviewer 1

Comments and Suggestions for Authors

Castro-Ribeiro et al study the cytotoxic effect of RGDS in GBM cells. Before the manuscript is suggested for publication in the journal, some concerns need to be addressed.

It is needed to determine whether RGDS-HA can induce apoptosis in GBM under conditions of MMP2 knockdown or the use of MMP2 inhibitors.

We acknowledge the suggestion of the reviewer. However, considering that MMP2 is only one of the proteolytic enzymes released from GBML42 cells that can disrupt the amide bond between RGDS and HA, as mentioned in the Results and Discussion sections of the manuscript (pages 11 and 15-16, lines 318-321 and 479-482, respectively), even with its knockdown or inhibition, the RGDS-HA induction of apoptosis may continue to be observed. To clarify that not only MMP2 can break the covalent bond between RGDS and HA, the following information was added in the abstract (page 1, lines 27-28):

“As GBM cells produce several proteolytic enzymes capable of disrupting the peptide-HA bond, we selected MMP-2 to illustrate this phenomenon.”

All abbreviations should be written out in full on first use. For example, PDI and DPPC.

We acknowledge the suggestion of the reviewer. We confirmed that all abbreviations were written out in full on first use. For instance, the names of the abbreviations PDI (page 3, line 117) and DPPC (page 3, lines 117-118) were included on their first use in the text.

When referring to concentration, "lower" is the more appropriate term than "smaller."

We agree with the comment of the reviewer. The word “smaller” was replaced by the word “lower” on pages 7, 10, and 15.

Comments on the Quality of English Language

Extensive editing of English language required.

The manuscript was revised to improve the English language.

Reviewer 2 Report

Comments and Suggestions for Authors

In this research study, the authors produced a hydrogel using hyaluronic acid (HMW-HA) and a fibronectin
peptide called RGDS, which has inhibitory properties. They found that the RGDS-hydrogel alone had a
cytotoxic effect on a GBM cell line derived from a patient. Additionally, the authors included large vesicles
(LUVs) in the hydrogel, which were used as physical crosslinkers and to contain the chemotherapeutic
agent doxorubicin (DOX). DOX is considered one of the most effective agents for treating cancer, but it has
limited efficacy in the traditional treatment of gliomas due to its low permeability across the blood-brain
barrier.
The study also found that RGDS was not cytotoxic in solution for cancer cells. However, the bond between
HA and peptides can be cleaved by proteolytic enzymes, such as MMP-2, released by tumor cells leading to
cancer cell death.
Finally, the researchers determined that the hydrogel formula containing LUVs+ DOX was toxic to tumor
cells but not astrocytes in a co-culture system with GBM cells, remaining viable for up to 10 days.
I found the article very interesting, as the use of natural polymers to modify TME signaling may represent a
valid strategy, as it addresses another characteristic that makes GBM such an aggressive tumor.
Moreover, a drug delivery system that can “release” the drug gradually over time at the specific site of
treatment, effectively reducing the toxic effects of traditional systemic chemotherapy administration, or
allowing the use of more effective agents that cannot cross the blood-brain barrier is a promising approach
for treating glioblastomas. Currently, these types of tumors have a very low survival rate and studies in this
direction could potentially improve patient prognosis.
After reviewing the article, I find that the experiments are well-designed and the text is clear, flowing, and
accurate in its bibliography. Therefore, I recommend publishing the article in the International Journal of
Molecular Sciences with minor revisions.
Minor suggestions:
- FIG. 9 - Has the release of RDGS from the functionalized hydrogel in the presence of MMP-2
significantly increased at 48 and 72 hours? If not, longer time points should be evaluated.
- FIG. 6 and supplementary figure S9 and the main text of "Hydrogel safety for astrocytes" paragraph
I found the paragraph on "Hydrogel safety for astrocytes" a bit confusing.
Both FIG.6 and S9 legends described the same experiment of GBML42 cells seeded with or without
150 μM LUVs, encapsulating 0.1 μM DOX or not. However, the metabolic activity graphs and
viability images are significantly different from each other.
Is Figure S9 referring to co-cultured cells mentioned in the main text?
At line 327, it was stated that the concentration of DOX was decreased due to the combined effect
of DOX and RGDS in killing GBML42 cells. However, it is not clear in which experiment the reduction
in DOX concentration occurred. Can you please clarify this point?
FIG. 10 refers to astrocytes seeded alone, correct?

Comments on the Quality of English Language

In this research study, the authors produced a hydrogel using hyaluronic acid (HMW-HA) and a fibronectin
peptide called RGDS, which has inhibitory properties. They found that the RGDS-hydrogel alone had a
cytotoxic effect on a GBM cell line derived from a patient. Additionally, the authors included large vesicles
(LUVs) in the hydrogel, which were used as physical crosslinkers and to contain the chemotherapeutic
agent doxorubicin (DOX). DOX is considered one of the most effective agents for treating cancer, but it has
limited efficacy in the traditional treatment of gliomas due to its low permeability across the blood-brain
barrier.
The study also found that RGDS was not cytotoxic in solution for cancer cells. However, the bond between
HA and peptides can be cleaved by proteolytic enzymes, such as MMP-2, released by tumor cells leading to
cancer cell death.
Finally, the researchers determined that the hydrogel formula containing LUVs+ DOX was toxic to tumor
cells but not astrocytes in a co-culture system with GBM cells, remaining viable for up to 10 days.
I found the article very interesting, as the use of natural polymers to modify TME signaling may represent a
valid strategy, as it addresses another characteristic that makes GBM such an aggressive tumor.
Moreover, a drug delivery system that can “release” the drug gradually over time at the specific site of
treatment, effectively reducing the toxic effects of traditional systemic chemotherapy administration, or
allowing the use of more effective agents that cannot cross the blood-brain barrier is a promising approach
for treating glioblastomas. Currently, these types of tumors have a very low survival rate and studies in this
direction could potentially improve patient prognosis.
After reviewing the article, I find that the experiments are well-designed and the text is clear, flowing, and
accurate in its bibliography. Therefore, I recommend publishing the article in the International Journal of
Molecular Sciences with minor revisions.
Minor suggestions:
- FIG. 9 - Has the release of RDGS from the functionalized hydrogel in the presence of MMP-2
significantly increased at 48 and 72 hours? If not, longer time points should be evaluated.
- FIG. 6 and supplementary figure S9 and the main text of "Hydrogel safety for astrocytes" paragraph
I found the paragraph on "Hydrogel safety for astrocytes" a bit confusing.
Both FIG.6 and S9 legends described the same experiment of GBML42 cells seeded with or without
150 μM LUVs, encapsulating 0.1 μM DOX or not. However, the metabolic activity graphs and
viability images are significantly different from each other.
Is Figure S9 referring to co-cultured cells mentioned in the main text?
At line 327, it was stated that the concentration of DOX was decreased due to the combined effect
of DOX and RGDS in killing GBML42 cells. However, it is not clear in which experiment the reduction
in DOX concentration occurred. Can you please clarify this point?
FIG. 10 refers to astrocytes seeded alone, correct?

Author Response

Author's Reply to the Reviewer 2 Report:

We kindly acknowledge and appreciate the comments and suggestions of Reviewer 2 that enabled us to improve and clarify our manuscript. The updated version of the manuscript ijms-2951342_Revised.doc includes the modifications based on the comments and suggestions proposed by the Reviewer, which were all considered and discussed in detail. In this updated version of the manuscript, the modifications proposed are marked in a different color.

Reviewer 2

Comments and Suggestions for Authors

In this research study, the authors produced a hydrogel using hyaluronic acid (HMW-HA) and a fibronectin peptide called RGDS, which has inhibitory properties. They found that the RGDS-hydrogel alone had a cytotoxic effect on a GBM cell line derived from a patient. Additionally, the authors included large vesicles (LUVs) in the hydrogel, which were used as physical crosslinkers and to contain the chemotherapeutic agent doxorubicin (DOX). DOX is considered one of the most effective agents for treating cancer, but it has limited efficacy in the traditional treatment of gliomas due to its low permeability across the blood-brain barrier.
The study also found that RGDS was not cytotoxic in solution for cancer cells. However, the bond between HA and peptides can be cleaved by proteolytic enzymes, such as MMP-2, released by tumor cells leading to cancer cell death. Finally, the researchers determined that the hydrogel formula containing LUVs+ DOX was toxic to tumor cells but not astrocytes in a co-culture system with GBM cells, remaining viable for up to 10 days. I found the article very interesting, as the use of natural polymers to modify TME signaling may represent a valid strategy, as it addresses another characteristic that makes GBM such an aggressive tumor. Moreover, a drug delivery system that can “release” the drug gradually over time at the specific site of treatment, effectively reducing the toxic effects of traditional systemic chemotherapy administration, or allowing the use of more effective agents that cannot cross the blood-brain barrier is a promising approach for treating glioblastomas. Currently, these types of tumors have a very low survival rate and studies in this direction could potentially improve patient prognosis. After reviewing the article, I find that the experiments are well-designed and the text is clear, flowing, and accurate in its bibliography. Therefore, I recommend publishing the article in the International Journal of Molecular Sciences with minor revisions.

Minor suggestions:

- FIG. 9 - Has the release of RDGS from the functionalized hydrogel in the presence of MMP-2 significantly increased at 48 and 72 hours? If not, longer time points should be evaluated.

We acknowledge the suggestion of the reviewer. At 24 h, approximately 22% of RGDS was released from the hydrogel. At day 3, this value was duplicated and it was significantly different from the amount released at 24 h. Figure 9, on page 12, was modified to include this significant difference:

Figure 9. RGDS release (%) from the functionalized hydrogel in the presence of MMP-2 for 24, 48, and 72 h. The symbol (*) denotes significant differences versus the 24 h time point: *p < 0.05.

- FIG. 6 and supplementary figure S9 and the main text of "Hydrogel safety for astrocytes" paragraph
I found the paragraph on "Hydrogel safety for astrocytes" a bit confusing.

The paragraph on “Hydrogel safety for astrocytes” on page 12, lines 345-358, was rephrased to improve clarity:

“The toxicity of the developed formulation for healthy cells, namely astrocytes (human immortalized astrocytes hTERT/E6/E7 cell line), was also evaluated. In this assay, the concentration of DOX was reduced to 0.1 µM. This reduction was performed considering the IC50 values of DOX at 2, 3, and 7 days and its synergistic action with RGDS in killing GBML42 cells. To confirm the effectiveness of this formulation, GBM cells were seeded, as in the previous assays, on the hydrogel. As can be observed in Figure S9A, the metabolic activity of GBML42 cells decreased after their culture on the hydrogel. These results were confirmed by the live/dead staining that showed a decrease in GBML42 cells’ viability (Figure S9B).

The metabolic activity of hTERT/E6/E7 increased (Figure 10A) and the live/dead results also confirmed that astrocytes remained viable in all time points tested (Figure 10B). These results indicate that the developed strategy is not cytotoxic for astrocytes.”

Both FIG.6 and S9 legends described the same experiment of GBML42 cells seeded with or without 150 μM LUVs, encapsulating 0.1 μM DOX or not. However, the metabolic activity graphs and viability images are significantly different from each other.
Is Figure S9 referring to co-cultured cells mentioned in the main text?

We appreciate the reviewer's comment that enabled us to correct a mistake in the manuscript. In Fig. 6 the concentration used of DOX was 4 µM, as mentioned in section 4.8.4. Efficacy of hydrogels in damaging GBML42 cells and not 0.1 µM, as presented by mistake in the legend. Additionally, the results presented in Fig. S9 correspond to the metabolic activity and viability of GBML42 cells that, while maintaining the relationship between cell number and area, were seeded on hydrogels placed in 24-well plates, and not in 48-well plates (as mentioned in the section 4.8.4) to enable for using inserts for the astrocytes cell line co-culture. Thus, the results obtained for cells’ metabolic activity and viability in the two experiments might not be directly comparable due to differences in, e.g., media volume that was used in the two conditions. However, both assays demonstrated that the functionalized hydrogel containing LUVs incorporating DOX is efficient in killing the cancer cells and, simultaneously, considering Fig. 10, is a safe approach for astrocytes.

Furthermore, a mistake was noticed in Figure S9B, where the control used was incorrect (HA) and was replaced by the correct control, corresponding to cells cultured on the well. This control was performed to verify the morphology and viability of cells during the 10 days of the assay. The control HA was not performed in this assay since, from Figure 6, it was already verified that HA is non-cytotoxic to GBML42 cells. The updated Figure S9 (supplementary information, page 8) is presented below:

Figure S9. Metabolic activity (A) and fluorescence microscopy images of live/dead staining with Calcein AM/PI (green: live cell; red: dead cell; B) of GBML42 cells cultured on 24-well plates (control: Ctr) or seeded on RGDS-functionalized HA (RGDS-HA) without or with 150 µM LUVs (RGDS-HA+LUVs) not or encapsulating 0.1 µM DOX (RGDS-HA+LUVs+DOX). In A, the symbol (*) denotes significant differences versus the Ctr: ****p < 0.0001; ***p < 0.001. The live/dead images (B) indicate the amount of live cells (green) in comparison to dead cells (red).

At line 327, it was stated that the concentration of DOX was decreased due to the combined effect of DOX and RGDS in killing GBML42 cells. However, it is not clear in which experiment the reduction in DOX concentration occurred. Can you please clarify this point?

The reduction of DOX was performed in the co-culture assay (Figure 10 and Figure S9), as previously referred. Thus, the paragraph of section 2.3.5, page 12, was changed to clarify this point, as indicated in the first answer.

FIG. 10 refers to astrocytes seeded alone, correct?

Figure 10 corresponds to the metabolic activity and viability of astrocytes seeded alone in an insert but in co-culture with GBML42 cells. The description of this assay is presented in section 4.8.7. Hydrogel efficacy and safety in a co-culture of GBML42 cells and astrocytes (page 20):

“RGDS-functionalized HA hydrogels at 5% (w/v) with and without LUVs encapsulating or not 0.1 µM DOX were transferred to 24 well-plates (200 µL per well). GBML42 cells (8×104 cells/well) were seeded on the top of the hydrogels. Afterward, the human immortalized astrocytes hTERT/E6/E7 cell line was seeded (1.5 ×105 cells/well) in 24 well-plate inserts with a 0.4 µm of pore size. Then, they were placed in each well containing the hydrogels seeded with GBML42 cells on their top. After cell culture (section 2.8.1) for 1, 3, 7, and 10 days, the inserts were removed, and metabolism and viability of the GBM cells and astrocytes were assessed separately, using MTS and live/dead assays, respectively, as described before.”

Reviewer 3 Report

Comments and Suggestions for Authors

An interesting approach to treatment of GBM is presented, with fairly comprehensive in vitro proof-of-concept studies. Experimental design is good on the whole, although I think a major flaw is the use of only one GBM cell line in the study and also the limitation of incorporating just doxorubicin into the hydrogel. I agree that doxorubicin is an effective anticancer drug, but it has it's own issues with resistance, and I think demonstrating  hydrogel incorporation and differential activity of another anticancer agent would strengthen the story significantly. Also the use of more sophisticated 3D in vitro models should be considered before the move to in vivo studies. Saying this, I think the submission almost stands on its own, and think with the addition of in vitro data using another cell line would merit publication.

Comments on the Quality of English Language

There are several cases of poor sentence construction which need to be addressed before publication.

Author Response

Author's Reply to the Reviewer 3 Report:

We kindly acknowledge and appreciate the comments and suggestions of Reviewer 3 that enabled us to improve and clarify our manuscript. The updated version of the manuscript ijms-2951342_Revised.doc includes the modifications based on the comments and suggestions proposed by the Reviewer, which were all considered and discussed in detail. In this updated version of the manuscript, the modifications proposed are marked in a different color.

Reviewer 3

Comments and Suggestions for Authors

An interesting approach to treatment of GBM is presented, with fairly comprehensive in vitro proof-of-concept studies. Experimental design is good on the whole, although I think a major flaw is the use of only one GBM cell line in the study and also the limitation of incorporating just doxorubicin into the hydrogel. I agree that doxorubicin is an effective anticancer drug, but it has its own issues with resistance, and I think demonstrating hydrogel incorporation and differential activity of another anticancer agent would strengthen the story significantly. Also the use of more sophisticated 3D in vitro models should be considered before the move to in vivo studies. Saying this, I think the submission almost stands on its own, and think with the addition of in vitro data using another cell line would merit publication.

The authors acknowledge the insightful inputs of the reviewer. We agree that doxorubicin (DOX), while effective, presents challenges related to resistance. Nonetheless, the synergic effect of DOX and RGDS proved to be efficient in damaging GBML42 cells during 10 days in in vitro assays (Figure 6 and Figure S9). Moreover, if DOX resistance is observed by cancer cells, we can easily encapsulate another drug to obtain high efficacy. The following sentence was added in the Discussion section (page 13, lines 371-375) to demonstrate the versatility of the developed formulation:

“However, if DOX resistance is observed, other anticancer drugs (e.g., prazosin, cerulenin, or orlistat) can be encapsulated in the hydrogel. Indeed, the versatility of the developed formulation will enable in the future the production of personalized treatments based on the patient GBM fingerprint.”

In this study, our main goal was to prepare and characterize the hydrogel and demonstrate its effectiveness in GBM cell killing. For this purpose, we conducted an extensive array of assays to characterize and demonstrate the efficacy of the developed hydrogel (e.g., we demonstrated that MMP2, a proteolytic enzyme expressed by GBML42 cells can release RGDS from the hydrogel). For proof-of-concept regarding the in vitro efficacy of the RGDS-functionalized hydrogel, we used GBML42, a human patient-derived primary glioblastoma (GBM) cell culture since it was established by our collaborator Bruno Costa a few years ago. Consequently, these GBM cells have not been in culture for as many years as the commercially available cell lines (e.g., U87 was established in 1966). This information was added in the Introduction section (page 3, lines 106-110). However, we agree with the reviewer that more assays are needed to demonstrate the efficacy and mode of action of the developed hydrogel. Indeed, the use of more sophisticated 3D in vitro models is an excellent suggestion to ensure more clinically relevant data and to better simulate the tumor microenvironment before progressing to in vivo studies. Therefore, this will also be considered in future assays, as mentioned in the manuscript (page 16, lines 506-508):

“Future assays will include the assessment of the hydrogel efficacy in different GBM cells and neural cells, as well as in more complex 3D cell models, before being tested in vivo.”

Comments on the Quality of English Language

There are several cases of poor sentence construction which need to be addressed before publication.

The authors agree with this comment and the text was edited to improve the English language.

Reviewer 4 Report

Comments and Suggestions for Authors

This manuscript provides extensive investigation of hyaluronic acid gels containing RGDS and liposomes encapsulating doxorunicin for glioblastoma treatments. In vitro characterization of hydrogels are comprehensive and the effects of RGDS –functionalization and incorporation of liposomes encapsulating DOC are clearly demonstrated. The manuscript requires minor editing, for example, degree symbols are underlined. I only have one question: hyaluronic acid crosslinking was not performed in this study, hence it is possible that gels formed are solubilized during tissue culture experiments. Did you observe solubilization of gels or 5% gel is physically crosslinked tightly?

Author Response

Author's Reply to the Reviewer 4 Report:

We kindly acknowledge and appreciate the comments and suggestions of Reviewer 4 that enabled us to improve and clarify our manuscript. The updated version of the manuscript ijms-2951342_Revised.doc includes the modifications based on the comments and suggestions proposed by the Reviewer, which were all considered and discussed in detail. In this updated version of the manuscript, the modifications proposed are marked in a different color.

Reviewer 4

Comments and Suggestions for Authors

This manuscript provides extensive investigation of hyaluronic acid gels containing RGDS and liposomes encapsulating doxorunicin for glioblastoma treatments. In vitro characterization of hydrogels are comprehensive and the effects of RGDS –functionalization and incorporation of liposomes encapsulating DOC are clearly demonstrated. The manuscript requires minor editing, for example, degree symbols are underlined. I only have one question: hyaluronic acid crosslinking was not performed in this study, hence it is possible that gels formed are solubilized during tissue culture experiments. Did you observe solubilization of gels or 5% gel is physically crosslinked tightly?

The authors acknowledge the comments of the reviewer. The text was edited to remove, for example, the underline of the degree symbols. Regarding the reviewer’s question, hyaluronic acid (HA) was not chemically crosslinked to not alter its properties. It was utilized as a component of the extracellular matrix. Throughout the 10-day assay, we observed some solubilisation of the HA in vitro. Indeed, after this period we could not separate the hydrogel from the culture medium. However, this is the intended behavior after implantation since we expect to have a hydrogel that can be easily permeated by the GBM cells that are in its surroundings. This will allow for the effective killing of the cancer cells.

Round 2

Reviewer 1 Report

Comments and Suggestions for Authors

The authors now have a much revised and improved paper. I have no further recommendations.

Comments on the Quality of English Language

 Minor editing of English language required.